

# Seismic and geologic controls on spatial clustering of landslides in three large earthquakes

Claire Rault[1], Alexandra Robert[2], Odin Marc[3], Niels Hovius[4, 5], Patrick Meunier[1]

[1] Laboratoire de Géologie, Ecole Normale Supérieure Paris, Paris, 75005, France
[2] Géosciences Environnement Toulouse, Observatoire Midi-Pyrénées, Toulouse, 31400, France
[3] Ecole et Observatoire des Sciences de la Terre, Strasbourg, 67084, France
[4] GFZ German Research Center for Geosciences, Potsdam, 14473, Germany
[5] Institute for Earth and Environmental Sciences, University of Potsdam, 14476, Germany

*Correspondence to*: Claire Rault (claire.rault@ens.fr)

**Abstract.** The large, shallow earthquakes at Northridge, California (1994), Chi-Chi, Taiwan (1999) and Wenchuan, China (2008) each triggered thousands of landslides. We have determined the position of these landslides along hillslopes, normalizing for statistical bias. The landslide patterns have a co-seismic signature, with clustering at ridge crests and slope toes. A cross check against rainfall-induced landslide inventories confirms that crest-clustering is specific to seismic-triggering. In our three study areas, seismic ground motion parameters, and lithologic and topographic features have limited bearing on the observed patterns of landslide clustering. However, we show that at the scale of the epicentral area, crest- and toe-clustering occur in areas with specific geological features. Toe-clustering of seismically-induced landslides tends to occur along major faults. Crest-clustering is concentrated at sites where the lithology along hillslopes is approximately uniform, or made of alternating soft and hard strata, and without strong overprint of geological structures. Although earthquake-induced landslides locate higher on hillslopes in a statistically significant way, geological features strongly modulate the landslide position along the hillslopes. As a result the observation of landslide clustering on topographic ridges cannot be used directly as an indicator of seismic parameters such as ground shaking.

## 1 Introduction

Seismic ground shaking triggers many landslides in active mountain areas. A growing number of catalogues of landslides associated with large earthquakes is now being produced by mapping from satellite images (Tanyaş et al., 2017; U.S. Geological Survey, 2018b). Such catalogues have been used to show that to the first order, the density of the co-seismic landslides is controlled by the intensity of seismic shaking and by hillslope rock strength, and that the total volume of landslides and the area extent affected by them increase with the earthquake magnitude (Keefer, 1984; Marc et al., 2016, 2017; Rodríguez et al., 1999).

In recent years, several studies have explored the position of the landslides in the landscape, adding this characteristic to their description of the landslide inventories. At the catchment scale, landslides triggered by storms and earthquakes affect different



parts of ridge and valley topography (Densmore and Hovius, 2000; Meunier et al., 2008). Storm-induced landslides are preferentially triggered low on slopes due to riverbank erosion and high groundwater pressure (Lin et al., 2011; Meunier et al., 2008; Tseng et al., 2017). By contrast, earthquake-triggered landslides are more uniformly distributed since ground shaking affects all portions of the hillslope (Densmore and Hovius, 2000), or they are concentrated near ridges or slope breaks (Harp and Jibson, 1996; Massey et al., 2017; Sepúlveda et al., 2010; Weissel and Stark, 2001). Numerical simulations of ground shaking in complex topographies predict that seismic waves are actually amplified around ridge crests (e.g. Boore, 1973; Massa et al., 2014; Poursartip et al., 2017). Both seismic noise analysis and strong motion records confirm that stronger shaking often occurs at topographic highs (Chávez-García et al., 1996; Durante et al., 2017; Hartzell et al., 2014; Massa et al., 2010). Meunier et al, 2008 suggested that earthquake-induced landslides tend to cluster around ridge crests as a consequence of these topographic site effects. Yet, amplification of ground shaking around the crests predicted by numerical studies is found to be modest, mostly 1,2 to 2.5 times the flat model, depending in particular on the shape of the hill and the seismic wave frequency considered (Ashford et al., 1997; Asimaki and Mohammadi, 2018; Chávez-García et al., 1996; Geli et al., 1988; Lovati et al., 2011; Pedersen et al., 1994). Numerous authors argued that larger crest amplifications observed in the field are mostly caused by lithological contrasts along the wave path and possible upward propagation of Rayleigh waves from the base of a slope towards the crest (Burjánek et al., 2014; Gallipoli et al., 2013; Glinsky and Bertrand, 2017; Havenith et al., 2003; Ohtsuki and Harumi, 1983).

Here, we study the spatial variations of the position of co-seismic landslides on hillslopes within the epicentral areas of three large, shallow earthquakes affecting steep mountain topography: the 1994 $M_w$ 6.7 Northridge Earthquake, the 1999 $M_w$ 7.6 Chi-Chi Earthquake and the 2008 $M_w$ 7.9 Wenchuan Earthquake. We also consider the location of rainfall-triggered landslides in the area affected by the Chi-Chi earthquake for comparison. Using a statistical approach, we identify coherent patterns of ridge crest- and slope toe-clustering. We explore seismic, topographic, lithological and structural features as possible controls on the observed patterns, and conclude that co-seismic landslide distributions are best explained by superposition of effects of local geological configurations on general seismically-controlled patterns.

## 2 Study areas and landslides inventories

We use previously published landslides inventories for three earthquakes (Table 1), constructed by digitizing landslides outlines from field and air-photos, and satellites images. These inventories have been shown to be relatively complete for landslides larger than 30 m², but they do not allow distinction between the erosional and depositional parts of landslides.

### 2.1 The 1994 Northridge earthquake

The $M_w$ 6.7 Northridge occurred on the 17[th] of January 1994, about 100km North of Los Angeles, in Southern California. Rupture started on the Northridge blind thrust fault, at about 19km depth (Somerville et al., 1996). It generated strong ground





shaking with peak ground accelerations (PGA) up to 1.78g. More than 11.000 landslides were triggered, with a cumulative area of more than 23km$^2$ (Harp and Jibson, 1996). Most of these landslides were located in the Santa Susanna and San Gabriel Mountains.

## 2.2 The 1999 Chi-Chi earthquake

On the 21$^{st}$ September 1999, the shallow $M_w$ 7.6 Chi-Chi earthquake occurred in the western foothills of Taiwan's Central Range. The rupture initiated along the Chelungpu thrust fault at 12±5km depth (Angelier et al., 2001). Strong ground shaking was recorded with a PGA up to 1g in some places. The earthquake caused about 10.000 landslides with a combined area in excess of 125km$^2$ (Liao and Lee, 2000).

## 2.3 The 2008 Wenchuan earthquake

On the 12$^{th}$ of May 2008, the $M_w$ 7.9 Wenchuan earthquake occurred along the eastern boundary of the Tibetan Plateau. The rupture initiated at a focal depth of 14 to 19km and propagated along two segments of the Longmen Shan thrust system (De Michele et al., 2010; Tong et al., 2010). Strong ground motion was felt with recorded PGA exceeding 0.8g in some places (Wen et al., 2010) . The earthquake caused a large number of landslides: more than 197.000 were mapped, with a cumulative surface area exceeding 700km² (Xu et al., 2014). At least three catalogues are available for this earthquake (Gorum et al., 2011;

Parker et al., 2011; Xu et al., 2014). Here, we use the one from Xu et al 2014b, which we deem to be the most complete and accurate, based on a comparison detailed in the supplementary materials.

## 2.4 Rainfall-induced landslides in the Chi-Chi epicentral area

In 2009, typhoon Morakot deposited up to.31.9 meter of rainwater in 48 hours in the considered area (Chien and Kuo, 2011). More than 15.000 landslides were triggered by this typhoon (Marc et al., 2018) in an area that extends into the Chi-Chi

epicentral area. In the area of overlap, the characteristics of the landslide populations associated with the earthquake and the typhoon can be compared directly. Moreover, we document the time variation of the location of the landslides located in three watersheds in the Chi-Chi epicentral area from 1994 to 2014 (Marc et al., 2015, Table 1). The location of these three catchments is reported in Fig. 2 and 3.



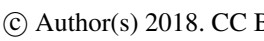

## 3 Methods

### 3.1 Landslide position in the landscape

Our aim is to determine the position of landslides within the landscape, specifically their position relative to a ridge crest or the top of a hillslope, and to the river valley or toe of a hillslope. For this, we adopt the metrics of Meunier et al, 2008 to normalize for the variation of hillslope lengths across the landscape, introducing the normalized distance to stream $|dst|$. Ridge crests are characterized by a $|dst|$ equal to one, while rivers have a $|dst|$ equal to zero (Method and metric Fig. S1e). Further,

we quantify the relative sampling of a given portion of the hillslope by landsliding by constructing the probability ratio $Rp$, that is the ratio of the probability density function of $|dst|$ of the cells covered by landsliding over that of all cells of the topography (Methods-metrics Fig. S1f).

### 3.2 Crest and toe clustering

To detect a possible external forcing in the position of landslides in the landscape, the observed distribution of landslides must

be compared to a distribution built from a random draw of landslide positions in the landscape (i.e. with no external forcing). The latter results from the geometry of the landscape itself plus or minus a random variation. Without external forcing, the expected value of $Rp$ in each interval $|d_{st}|$ should fall within a prediction interval associated with a random draw (Fig.S1f). For the purpose of comparison, we consider that the population of landslides resulting from an earthquake is equivalent to a single draw.

We define $Rp_{crest}$ as the average of $Rp$ values computed over the upper quarter of hillslope segments. Crest-clustering, defined here as preferential sampling of the upper quarter of a hillslope section by landsliding (cf. Meunier et al., 2008), is only considered to occur where $Rp_{crest}$ exceeds the upper bound of the 90% prediction interval associated with a random distribution $I_{rp}$ (see Methods-Statistics). This condition ensures that there is only a 10% chance for clustering to be due to a statistical bias. Similarly, $Rp_{toe}$ is defined as the average of $Rp$ computed over the lower quarter of the slope, and toe-clustering is adjudged

for $Rp_{toe}$ values greater than the upper bound of $I_{rp}$. Since crest-clustering and toe-clustering are mutually exclusive (see Methods-Statistics Fig. S3), zones of toe-clustering must have values of $Rp_{crest}$ lower than the lower bound of $I_{rp}$.

### 3.3 Spatial mapping of the landslide position within the epicentral area

Maps of $Rp_{crest}$ and $Rp_{toe}$ were generated by subdividing a study area into macrocells in which $Rp$ is calculated. The size of the macrocells in this study is set at 7.8 km$^2$ to optimize for two criteria: a) the cell must be small enough to capture the spatial

variation within the epicentral area, and b) it must be large enough to be statistically representative in terms of the number of landslides contained (see Methods-Statistics). The second criterion imposes a lower limit to the resolution at which we can





observe any spatial variation. Due to its size, a macrocell typically contains multiple individual slopes, even where the mountain relief reaches up to several kilometers.

## 3.4 Extraction of seismic and topographic parameters

In each macrocell, we compute the median of the seismic parameters according to the USGS ShakeMap (Allen and Wald 2007; U.S. Geological Survey 2018a). Shake maps provide the peak ground velocity (PGV), peak ground acceleration (PGA), and the pseudo spectral acceleration (PSA) at 3s, 1s, and 0.3s.

Ground motion can vary significantly along the topography due to site effects. Seismic amplification at the scale of the topography is approximated with the frequency scale curvature (FSC) method proposed by Maufroy et al., 2015. In this method

the median amplification factor (MAF) at frequency $f$ is equal to:

$$MAF(f) = 0.008 \times \lambda s \times Cs(Ls) + 1, \tag{1}$$

where $Cs(Ls)$ is the curvature of the topographic surface in a digital elevation model (DEM), smoothed over a length $L_S$ which is defined as $Ls=\lambda s/2$ with $\lambda s$ the S-wavelength considered. Because seismic wave velocity profiles are not well described in the study regions, the amplification is analyzed as a function of S-wave length instead of frequency. Maps are computed of the

maximum MAF for S-wave lengths ranging from 500 to 833m.

At larger scale, relations have been observed between seismic ground motion and the topographic shape and orientation with respect to the epicenter (e.g. Paolucci, 2002). To test for such effects, we simplify the topography considering triangular ridge cross-sections. To do this, we perform a geometric extraction of the ridge relief, ridge half-width and calculate the ridge shape ratio (ridge relief /ridge width) (see Extraction of topographic features Fig. S8).

## 3.5 Lithological features

In order constrain the influence of rock strength on landslide location patterns, we group lithologies that have similar apparent physical properties, using the information provided by geological maps of the earthquake epicentral areas (see Additional information on clustering controls Fig. S10). For the Northridge area, we use a combination of the maps compiled by the United States Geological Survey (Yerkes et al., 2005; Strand, 1969). For Taiwan we use materials from the Taiwan Central

Geological Survey, MOEA (MOEA and Central Geological Survey, 2008), and for Wenchuan we use the map published by Robert, 2011. Each macrocell is defined by its dominant lithology group, *i.e*, the one occupying the largest area.



## 4 Results

### 4.1 Temporal variation of crest-clustering

To test if seismic ground shaking and rainstorms cause hillslope failures in different parts of the landscape, we first consider

the temporal variation of clustering in the upper quartile of slopes in three watersheds in the Chi-Chi epicentral area between 1996 and 2014 (Fig. 1). Before the Chi-Chi earthquake, the typhoon-induced landslides tended to under-sample the upper slope domain ($Rp_{crest} < 0.6$). The Chi-Chi earthquake itself was characterized by clear crest over-sampling ($Rp_{crest} = 1.2$). Just after the earthquake, $Rp_{crest}$ dropped to 0.4 and returned to its pre-earthquake value in about 3 years. This evolution confirms that landslides triggered by earthquakes and rainfall have distinct and different clustering behaviour.

### 4.2 Spatial variation of crest-clustering

Figure 2 shows the spatial distribution of $Rp_{crest}$ in the three epicentral areas. Macrocells without statistically significant clustering are removed for clarity (see Methods-Statistics). In the three cases, we observe coherent patterns of crest- and toe-clustering on about half of the surface affected by landsliding (Fig. 2). These patterns can cover several tens of square kilometers, and they have similar sizes in the three epicentral areas. Hence, the larger the epicentral area the more individual

patterns we observe. Specifically, the Northridge epicentral area is almost exclusively affected by crest-clustering (Fig. 2b). Two coherent zones are observed in the Chi-Chi epicentral area: crest-clustering in the western-part of the epicentral area, and toe-clustering in the eastern part (Fig. 2c). In the Wenchuan case, five or six separate patches of crest-clustering can be identified. They are separated by more or less elongated zones of toe-clustering extending up to several tens of kilometers (Fig. 2a). Overall, crest-clustering does not appear to be a dominant pattern in the Wenchuan case, in contrast to the other two cases.

Note that in the Wenchuan case, the pattern of clustering is very sensitive to the quality of the landslide inventory (Fig. S6 and S7). In the following, we only consider results obtained with the Xu et al, 2014 dataset. Combined, the three cases shows that earthquake-triggered landslides are distributed quite evenly along many slopes in an epicentral area, with upper slope or slope toe clustering in some places.

The spatial distribution of $Rp_{crest}$ for the landslides induced by typhoon Morakot in Taiwan is distinct from that found for the

three earthquakes. The typhoon caused uniform toe-clustering (Fig. 3), with lower values of $Rp_{crest}$ (~0.5) in the aforementioned watersheds than those obtained in the same region for landslides induced by the Chi-Chi earthquake, even though these also cluster downslope. This observation suggests that toe-clustering is a signature of rainfall-induced landslides.

The patterns presented in Fig. 2 can be biased by the landslide mapping technique. The inventories we use do not distinguish between landslide deposits and scars. As landslides move downslope, by definition they preferentially affect the lower parts

of hillslopes. To test the robustness of our results, we have run the same analysis with the data from Taiwan, using the landslide centroids and estimated landslide scars. To do this, we  determined the length and the width of individual landslides, and used



the finding of Domej et al. (2017) that Earthquake-triggered landslides scars have a stable width to length ratio of *Ar=0.6*. The length of a landslide is equal to the difference between its maximum and minimum distance to river. The width is calculated using the landslide length and area, assuming a rectangular shape. Then, the lower part of the landslide polygon is progressively

removed until *Ar=0.6*. The Northridge and Wenchuan inventories contain too many instances of landslide amalgamation to perform a systematic, accurate scar extraction. For the Taiwanese case, the values of $Rp_{crest}$ obtained from centroids and scars are plotted against the values obtained using the whole landslides in supplementary Fig. S4. The results from these two methods have a nearly 1:1 correlation. Therefore, the regional pattern of $Rp_{crest}$ seems to be preserved, irrespective of whether we consider whole landslides or landslide scars.

## 5 Controls of clustering


Pervasive crest-clustering of co-seismic landslides within an earthquake epicentral area would signal predominance of seismic controls over any other controls. By contrast, a noisy distribution of crest-clustering could suggest that the location of landslides is controlled by highly variable local factors such as topographic slope, soil moisture or soil depth. The existence of patterns of crest-or toe-clustering over tens of square kilometers suggests a large-scale control such as regional geological structures or

geomorphic features.

### 5.1 Seismic controls

In our examples, crest-clustering is not primarily explained by regional seismic parameters. Figure 4 shows $Rp_{crest}$ plotted against the median of Peak Ground Velocity (*PGV*) (Fig. 4a) and Pseudo Static Acceleration at 1s (*PSA 1s*) (Fig. 4b). For the Northridge and Wenchuan earthquakes, crest- and toe- clustering both occur over a wide range of *PGV* (1-100 cm/s) and *PSA*

(0.1-1g). In Taiwan, $Rp_{crest}$ weakly increases with *PGV* and $PSA_{1s}$ but the spatial distribution of the patterns relative to the regional geological structure may cause misattribution. Indeed, as *PGV* and *PSA* strongly decrease towards the east, the strength of the geological units increases (see Sec. 5.3) (see Additional information on clustering controls Fig. S11). Similar results are found for *PGA* and $PSA_{3s}$ (see Additional information on clustering controls Fig. S9).

Figures 5 shows the crest-clustering of co-seismic landslides as a function of the median amplification factor *(MAF)* in the

Wenchuan case. No relation is observed at the scale of the entire epicentral area (Fig. 5a). Nevertheless, in some places, the landslides cluster at ridge crests where the predicted ground-motion amplification exceeds 1.8. This is particularly well expressed along ridges that are parallel to the fault system (Fig. 5b and 5c. and see Additional information on clustering controls Fig. S14). For example, in the Tangwanzhai syncline, half of the ridge crest-landslides are on area with *MAF* higher than 1.2 (Fig.5c). However this observation cannot be generalized to the whole epicentral area as there is no global correlation

between *MAF* and $Rp_{crest}$ (Fig. 5a).





## 5.2 Geomorphic controls

Local hillslope geometry does not explain cluster location. Figure 6c shows $Rp_{crest}$ plotted against the median of the ratio of the gradient of the upper and lower hillslope quarters. No correlation can be identified. Both hillslope local relief and aspect ratio also fail to segregate zones of crest-clustering from zones of toe-clustering (Fig. 6a-b) with possible exception of the Chi-Chi case. There $Rp_{crest}$ seems to decrease as slopes become higher and steeper.

## 5.3 Geological control on $Rp_{crest}$ distribution

Maps of $Rp_{crest}$ projected on the main lithological units of the three epicentral areas are shown in Fig. S10. Meanwhile, the statistical distributions of $Rp_{crest}$ per lithology are reported in boxplots in Fig. 7. In the Chi-Chi case, crest-clustering is principally observed in the western foothills that are comprising of poorly consolidated sandstones with interbedded marls and mudstones (Camanni et al., 2014; MOEA and Central Geological Survey, 2008) (Fig. 7b. and see Additional information on clustering controls Fig S10c). The higher grade lithological units to the east are mainly affected by toe-clustering. Hence, lithology seems to be a first-order control on the distribution patterns of $Rp_{crest}$.

However, in Northidge and Wenchuan cases, the distribution of $Rp_{crest}$ is not correlated in a simple way with rock strength according to simple lithological classes (Fig. 7a and c). In the Wenchuan epicentral area, rocks with various deformation grades are observed, depending among other things on the major geological structures that intersect them. For instance, intensely deformed sandstones are found into the Wenchuan Shear Zone, while they are moderately deformed within the Songpan Garze units, and relatively intact in the foothills (e.g. Robert, 2011). This geological diversity allows a more detailed, ad hoc analysis of substrate controls on landslide location.

From our data, crest- and toe-clustering of co-seismic landslides seem to be concentrated along specific geological features. This is illustrated by the following observations from the Wenchuan epicentral area, which is large and geologically diverse.

In the Wenchuan shear zone, landslide toe-clustering occurs along the Minjiang river valley (Fig. 8b). This river is entrained in the Wenchuan shear zone over more than 60km, where deformation of rocks is very intense (e.g. Godard et al., 2010; Liu-Zeng et al., 2011). In this area, mostly Paleozoic rocks have several schistosities and intense foliation that strongly decrease their strength (cf Fig.8 cross section A-B). The deformation is particularly intense in this zone due to the presence of resistant granitic massifs on both side of the fault zone (Robert, 2011). The most weakened material is downslope where the fault cuts the surface.

The central part of the foothills of the Longmen Shan is characterized by two large units: the so-called "upper unit" has large lithological contrasts over short distances (~10km) due to folding and thrusting, while the "lower unit" is more uniform (Fig.8a and c cross section C-D). In the upper unit co-seismic landslides have a coherent toe-clustering pattern, whereas the lower unit has a clear crest-clustering pattern (Fig. 8a and S12b). A strong concentration of landslides is observed on lower slope segments along the Beichuan fault, especially up to the Jinhe and Mianyuan rivers branches (Fig. 8 and S12b). Around this fault, massive





Permian dolomites top the cataclastic Triasic rocks, which crop out along the Tuojiang river, forcing failures downslope. In the area of Qiaping, between the Beichuan fault and the Pengguan massif and along the Mianyuan river, the Silurian and Devonian sedimentary rock layers are dipping steeply and bear traces of strong deformation, including pervasive schistosity as well as dissolution figures (Robert, 2011) (Fig. 8a and c cross section E-F). There, the downslope layers could be more susceptible to toppling onto the riverbed. The location of landslides is thus strongly controlled by the stratigraphy (weak rocks downslope topped by strong rock forming the crests), bedding dip, and the fault weakening zone.

In the foothills of the Longmen Shan, except the central part discussed above, crest-clustering of landslides is clearly dominant. In the north-eastern part, most of the landslides oversampled the crests of the large Tangwanzhai syncline (Fig. 8 and 5b). In this area, the presence of this large syncline strongly influences the morphology as the crests formed by sandstone and limestone strata are almost parallel to the Wenchuan fault system (Fig. 8a and c cross section G-H and Fig. S13 cross section I-J). Similar patterns are observed in the Sanjiang klippe and on both sides of the Tuojiang river, in the Longmen Shan Central Zone (Fig. 8, S12a and S13 cross section K-L). These crests are made of stronger and more resistant rocks implying the formation of steep slopes in the direction opposite to the dip of sedimentary layering. This slope asymmetry is marked by a strong curvature along the crests, a configuration that could favour amplification of ground-motion promoting toppling or wedge failures.

Finally, in earthquake affected Crystalline Massifs of the Longmen Shan (Pengguan, Xuelang Bao and Baoxing), crest-clustering is also dominant, except along the Minjiang river (Fig. 8a).

Meanwhile, in the Chi-Chi epicentral area, crest-clustering is observed in the foothills made of terrace deposits and alternating of sandstone and shale strata (Fig. S10c). Toe-clustering is found in the eastern part of the epicentral area where steep valleys and important fault system are cutting the shaly sandstones and slightly metamorphosed argilite layers (Fig. S10c).

In the Northridge area, crest-clustering is observed where interbedded conglomerate sandstones and shales form the crests of the Northern part of the Santa Susanna Mountains (Harp and Jibson, 1996; Winterer and Durham, 1962) (Fig. S14). There, co-seismic landslides preferentially occurred on the steepest slopes cutting across the stratigraphic dip (Fig. S14). This configuration seems to be similar to that in the Tangwanzhai area of the Wenchuan earthquake. Crest-clustering is also observed in the so called badlands at the fringe of the Santa Clarita basin which have formed in a homogeneous weak lithology (Fig. S10b).

In summary, three main types of geological effects were identified as major controls on landslide clustering: a) rivers flowing along fault zones with structurally weakened rocks, b) stratigraphic alternations of strong and weak units and c) the effect of the bedding on very steep hillslopes.

## 6 Discussion and conclusion

In this study we have systematically tested for a range of controls on the position of co-seismic landslides relative to the toe and the crest of hillslopes. Confirming previous studies (*e.g.* Densmore and Hovius, 2000, Meunier et al., 2008), we find that





rain-triggered landslides occur preferentially at slope toes, likely due to high pore pressures associated with infiltration and

260 fast downslope flow of groundwater in fractured rock mass, regolith and colluvium during rainfall. The location of earthquake-triggered landslides is, on average, higher on hillslopes than the rainfall-induced one, and displays coherent patterns of toe-and-crest clustering spread all over the epicentral area. Where we have identified clear patterns of crest- and toe-clustering within the epicentral area of Northridge, Chi-Chi, and Wenchuan earthquakes, these are due to a combination of seismic mechanisms and geological controls.

Toe-clustering of seismically-triggered landslides occurs mainly in areas where hillslope materials are heavily fractured, weathered, particularly in river valleys along major fault zones, and more specifically near the fault where the deformation is the highest. The influence of fault zone weakening on slope stability have been documented in other contexts (e.g. Demir et al., 2013; Korup, 2004; Scheingross et al., 2013). In absence of particular geological structures toe-clustering is also observed along trunk valleys in hard rock massifs where static stress can have induced severe fracturing at the base of topographic ridges

(Molnar, 2004), and where weak stratigraphic units crop out low in mountain landscapes. Therefore toe-clustering of co-seismic landslides appears to be explained at the first order by geological and structural controls. These controls add to any effects of possible downslope seismic amplification due to surface wave generation or directional effects (Pilz et al., 2018; Wasowski et al., 2013).

Crest-clustering of co-seismic landslides is found primarily is areas without strong lithological contrasts,specific geological

structures or away from river trunk valleys. It is particularly well developed in regions underlain by sedimentary rocks, where ridge crests are defined by specific beds oriented parallel to the seismogenic faults. In these particular geological configurations, ground-shaking could control the landslide position as it may be revealed by the correlation of crest-clustering with the MAF in the Tangwanzhai syncline (Fig 5a). The landslide position would thus reflect of the expression of strong ground motion in the uppermost part of the slope which can be explained by complex interactions of various seismic waves

with both topography and lithology. The focusing of waves on the edges of slopes may induce sufficient amplification of the ground motion to trigger slope failures (e.g Kaiser et al., 2013; Stahl et al., 2014). Higher levels of amplification may be reached when the incoming wave is perpendicular to the ridge elongation (Massa et al., 2010), and thus increase the probability of failure. Moreover some authors suggest that Rayleigh waves, generated at the toe of the hillslope and propagating toward the ridge-crest, would produce an added inertial force on the sliding mass and increase the duration of ground motion, favoring

upper slope failures (Jafarzadeh et al., 2015; Poursartip and Kallivokas, 2018).

We do not find clear explanations for the presence of some of the large crest- and toe- clustering patterns, as in Wenchuan along the Subo river, or east of Beichuan. Additional field observations in these area may help to document these signals.

Our results reconsider the hypothesis of Meunier et al, 2008 since we show that the co-seismic landslide position along hillslopes is strongly modulated by geological features (stratigraphy and bedding) and structures (faults and folds). The ground

motion intensity controls the landslide density (Meunier et al., 2007; Yuan et al., 2013), and seems to influence the distribution of the landslide size (Marc et al., 2016, Valagussa et al., accepted), but seems to be a secondary control on their positions along hillslopes in geologically contrasted epicentral areas. Therefore, hazard scenarios for earthquake-induced landslides should



not consider only lithology-units but strive to also consider stratigraphic and structural objects that can favor landsliding on specific hillslope sections.

**Author contributions**

C. Rault has developed the method computed results and prepared the manuscript.

P. Meunier has developed the method, computed results and prepared the manuscript.

A. Robert, has contributed her expertise on geological structures in the Wenchuan earthquake area and prepared the manuscript.

O. Marc has mapped the rainfall-triggered landslides in Taiwan and prepared the manuscript

N. Hovius has participated into of method development and preparation of the manuscript

**Competing interests**

The authors declare that they have no conflict of interest.

**Acknowledgments**

We thank M. Pubellier for its relevant discussions.

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




**FIGURES**

**Table 1: Details of the inventories used of earthquake- and rainfall-induced landslides**

| Database | Date | Country | Number of landslides | Surface covered by landslides (km²) | Trigger | Landslides inventory origin | Data and Methods used to map the landslides |
|---|---|---|---|---|---|---|---|
| **Pre-Chi-Chi** | 1994 - 1999 | Taïwan | 375 | 2.7 | Rainfall | (Marc et al., 2015) | Satellites images |
| **Chi-Chi** | 1999 | Taïwan | 9272 | 127.6 | Earthquake | (Liao & Lee, 2000) | Aerial photographs and satellites images |
| **Post-Chi-Chi** | 1999 - 2004 | Taïwan | 1647 | 10.1 | Rainfall | (Marc et al., 2015) | Satellites images and air photos |
| **Morakot** | 2009 | Taïwan | 17344 | 225.0 | Typhoon | (Marc et al., 2018) | Satellites images |
| **Wenchuan** | 2008 | China (Sichuan) | 197 481 | 1 160 | Earthquake | (Xu et al, 2014) | Aerial photographs and satellites images |
| **Northridge** | 1999 | USA (California) | 11 111 | 25.9 | Earthquake | (Harp & Jibson, 1996) | Air photos and field observations |

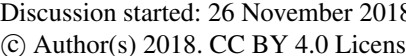
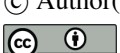
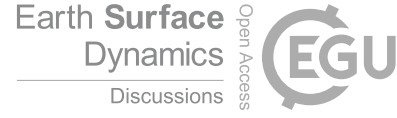

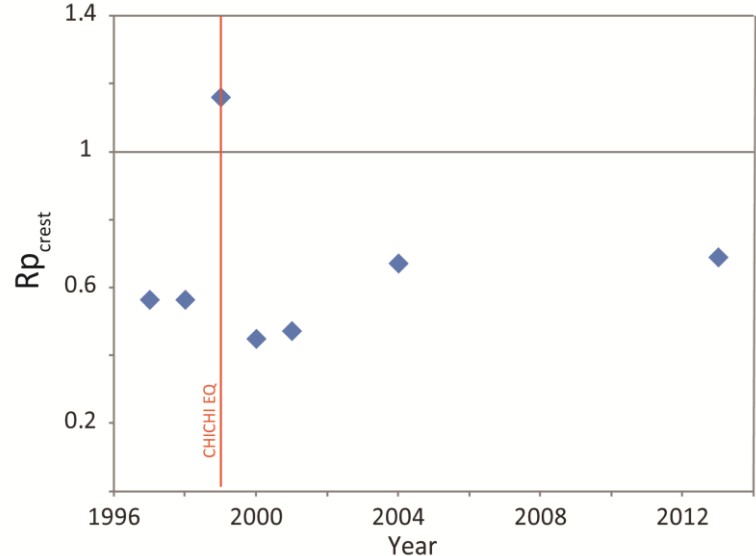

**Figure 1: Time variation of the landslide crest-clustering $Rp_{crest}$ in three watersheds in the Chi-Chi epicentral area mapped in Fig. 2. Chi-Chi-induced landslides sit well above the previous and subsequent rainfall-triggered landslides.**





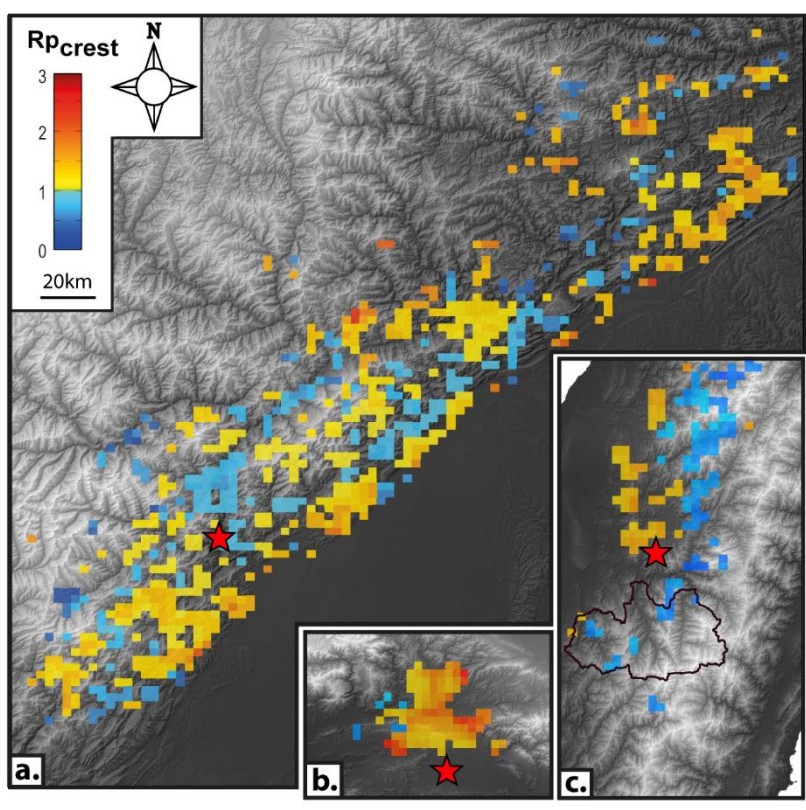

**Figure 2:** *Rp_crest* **maps in the a. Wenchuan, b. Northridge and c. Chi-Chi epicentral area. The 3 maps are at the same scale. The study area are divided in macrocells of 7.8km². Only cells of *Rp_crest* above the 90% prediction interval are represented (see Methods and Metrics). Regions of crest-clustering are colored in yellow-red. Regions of toe-clustering are colored in blue. Clear coherent patterns of crest- and toe-clustering are identified. The black curve delimits the 3 watersheds where *Rp_crest* is documented between 1996 and**

**2014 (Fig. 1).**



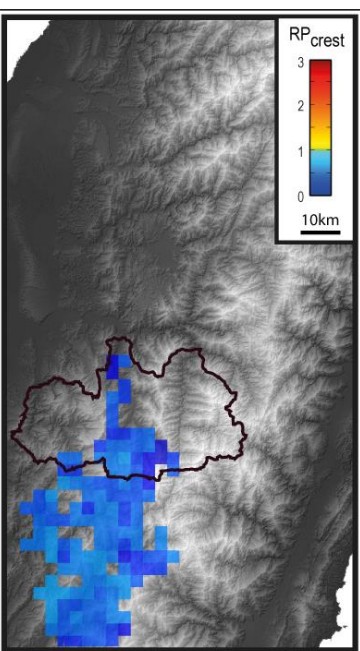

**Figure 3:** *Rp$_{crest}$* **map associated with the Morakot induced landslides in the southern west part of the Chi-Chi epicentral area. Only toe-clustering is observed. The black curve delimits the 3 watersheds where *Rp$_{crest}$* is documented from 1996 to 2014 (Fig. 2).**

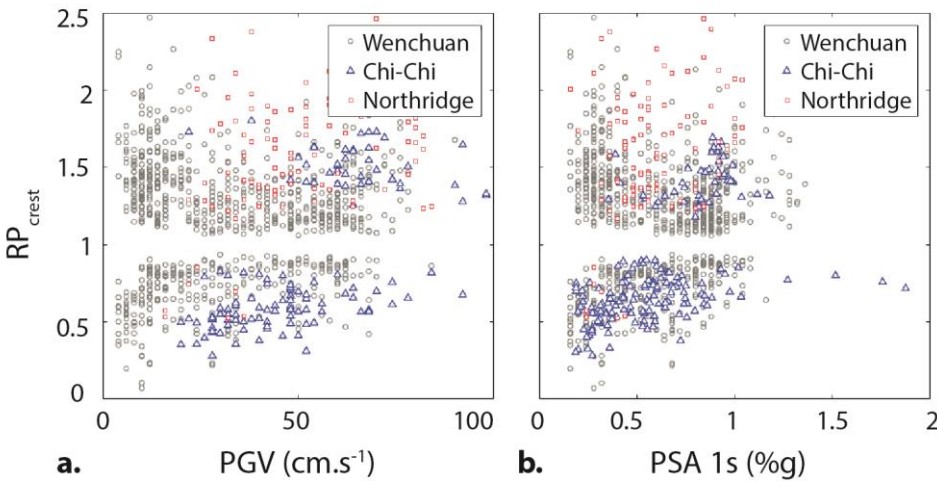

**Figure 4:** *Rp$_{crest}$* **as a function of seismic features: a. Median Peak Ground Velocity (PGV) (m.s$^{-1}$), b. Median Pseudo Spectral Acceleration at 1s (PSA 1s) calculated in the Wenchuan, Northridge and Chi-Chi epicentral areas. Regional seismic parameters do not seem to explain landslide position along hillslopes.**



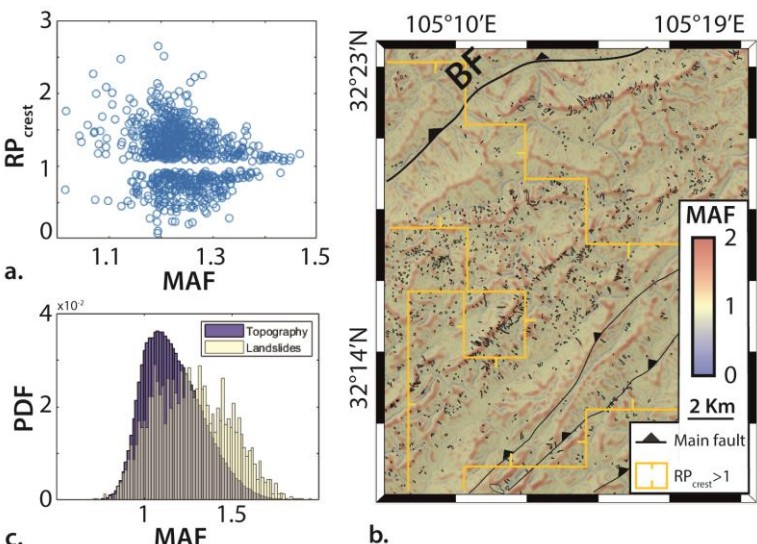

**Figure 5: Position of the landslides along hillslopes compared to the 80% percentile of the Maximum medium Amplification**
**Frequency scale curvature (*MAF*) over the S-wavelength interval of [500-833m] in the Sichuan. a. *Rp_crest* as a function of the *MAF*, no relation is observed. b. Map of the landslides (black polygons) superimposed on *the MAF* map in the Tangwanzhai nappe. The snapshot position is given in Fig. 8a. The yellow polygons surround macrocells with *Rp_crest* higher and lower than 1 respectively. c. Distribution of the *MAF* of the upper part of the hillslope (|dst|>0.7) in all the topography and in the 20% upper part of the landslides. Landslides tend to oversample areas with a *MAF*>1.3.**

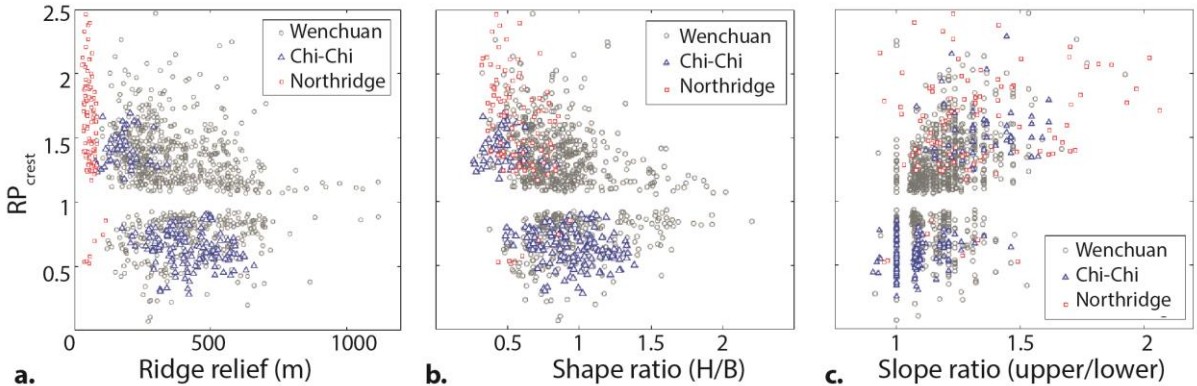

**Figure 6: *Rp_crest* as a function of topographic features: a. ridge relief, b. hill shape ratio (H ridge relief, B half width of the hill) and c. upper over lower hillslope gradient ratio calculated in the Wenchuan, Northridge and Chi-Chi epicentral areas.**



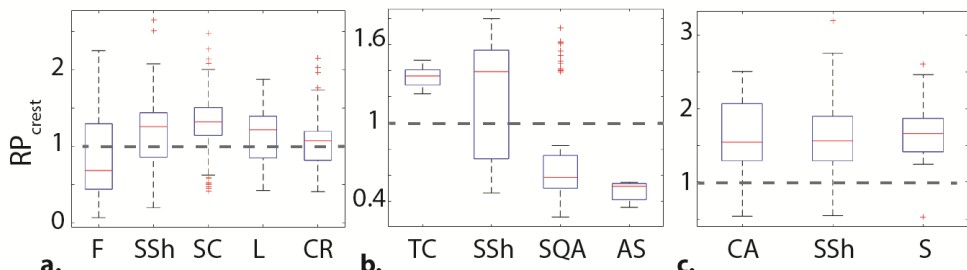

**Figure 7:** *Rp$_{crest}$* **as function of the lithologic groups of the a. Wenchuan, b. Chi-Chi and c. Northridge epicentral areas. F: flysh; SSh: mostly sandstones and shales; SC: mostly sandstones and conglomerates; L: mostly limestones; CR: crystalline rocks; TC: terrace deposits and conglomerates; SQA: shaly sandstones, quartzite and argillite; AS: argillite and sandstones; CA: colluvium and alluvium; S: mostly sandstones.**

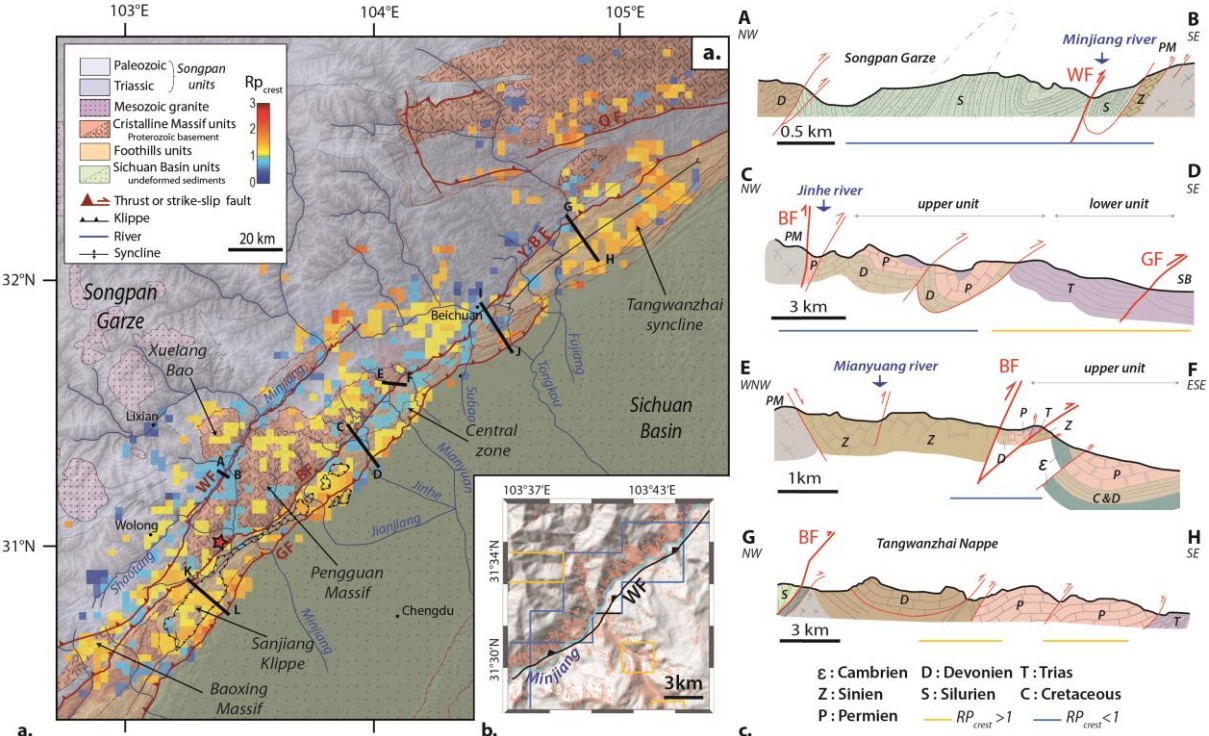

**Figure 8: a. Structural map of the Wenchuan earthquake epicentral area (after Robert, 2011) overlaid with the *Rp$_{crest}$* map. b. Snapshot of the landslide map in a portion of the Wenchuan shear zone. Its location is reported in Fig. 8a. Polygons with red contours represent the co-seismic landslides mapped by Xu et al., 2014. The yellow and blue lines delimit zones of crest- and toe- clustering respectively. Other snapshots of the co-seismic landslide maps are presented in Fig. 5.b, S12a and S12b respectively. c. Cross sections of different**

**structural units after Robert 2011. Cross sections I-J and K-L are presented in Fig. S13. GF: Guanxian fault, BF Beichuan fault, WF Wenchuan fault, Y-B F Yinxiu-Beichuan fault, QF Qinling fault.**