# Peer review of "Seismic and geologic controls on spatial clustering of landslides in three large earthquakes"

_Earth Surface Dynamics, 2018_

## Referee Comment (RC1) · Anonymous Referee #1 · 7 Mar 2019

In this paper, the authors explore ridge and toe clustering of earthquake-triggered landslides compared to rainfall-triggered landslides and seek explanations for different patterns, particularly related to geology. While, if taken at face value, their results are compelling and relevant, the problem is the paper and especially its supplement lack sufficient clarity for me to determine if their methods are scientifically sound and unbiased.

It was very difficult to follow the methods description, which is mainly detailed in the supplement, because of English usage problems and misspellings, but also lack of clarity and logical gaps that require more explanation. This may be partially because I don't specialize in probability and statistics, so it may be worth having someone with more expertise in that area review this instead of me if revised or resubmitted. But

even so, most readers are not going to be specialists in probability and statistics either yet they should still be able to follow the steps and logic of what was done. Therefore, I think the authors need to do a substantial rewrite of the methods section and supplement for clarity and have the supplement reviewed for English usage before this is given to another reviewer.

Without having fully understood the methods used by the authors to measure clustering, I do still have a few general concerns. The authors state several times that they confirmed rainfall-induced landslides cluster at toes and earthquake-induced slides often cluster at crests in general in their analysis, but this statement is made based on comparing against just one rainfall inventory. That is not enough evidence to make such a strong, generalized statement. I'm quite sure there are other rainfall induced landslide inventories out there, adding one or two more to the analysis would provide more support to the general statements made.

The implementation of the topographic amplification estimation method needs to be explained in much greater detail, for it's not clear what they actually did to compute MAF, especially given that the method is frequency-dependent but they make no mention of assumptions of S-wave velocities or how they dealt with the frequency-dependence or how they chose the wavelength range they mention. But from what I can understand, I'm not sure their exploration of MAF as an explanatory factor for ridge clustering is not really telling us much, given the huge uncertainties in estimating topographic amplification, that they average the results over the very large macrocells, and the fact that the method they use is based on numerical models and has not really been proven effective with real data yet. Note, there is an alternative empirical method that is not mentioned in the paper that might be worth comparing because in contrast to the Maufroy method, it is based on data. The downside is its specific to California data. Rai et al., 2016 https://doi.org/10.1193/113014EQS202M

The authors are also clearly aware that using the entire polygon of a landslide biases their results and they explore this. They do a small analysis of the centroids of the

source areas for the Chi-chi earthquake and state that it shows the same pattern, concluding that they can disregard the potential bias, but I'm not convinced they did a thorough job of ruling out bias because they derived the source areas from the original polygons automatically based on a simple assumption of typical aspect ratio and only looked at one inventory. I would be more convinced if they instead used one of the several inventories that did map source areas and deposits separately [e.g., Gorkha (Roback et al., 2017), Mid-Niigata (GSI of Japan, 2005), Kaikoura, (Massey et al., 2018)] to show whether considering entire polygons biases the results.

Line-specific comments

L13 – The word "confirms" is a little strong for a conclusion based on comparison against one rainfall inventory

L14-15 – Stating that seismic ground parameters have little bearing on observed patterns is pretty problematic because the landslides wouldn't have happened without the ground motion. The greater likelihood is that we don't have the means to accurately estimate the relevant ground motion parameters at the site where landslides are often triggered. Perhaps rather than saying "have little bearing" one could instead say ground motion parameters from ShakeMap do not seem to exert a primary control on observed clustering patterns.

L16-17 By major faults, do the authors mean faults involved in the earthquake that triggered the landslides or all faults?

L20-21 I don't think anyone is suggesting that landslide clustering be used as an indicator of seismic parameters. . .

L61 – give reference/source of reported PGA's here and elsewhere. Use commas for thousands in English, dots for decimals.

L95 – What does a random draw of landslide positions with no external forcing even mean? Pretty much every landslide occurs due to external forcing. Perhaps this section

needs to be rewritten for clarity?

L121 – How is the Maufroy method actually implemented? Not nearly enough detail is given. The method is frequency-specific, but the relevant frequency depends on the scale of the feature and the wavelength depends on the shear wave velocity. Also how is this applied to the ground motions, none of the ShakeMap outputs are frequency-specific except the spectral accelerations, but those are single degree of freedom oscillators with a specific natural frequency, which is not the same thing as ground motion of a specific frequency content. This method is also based on modeling results and to my knowledge, hasn't yet really been validated against real data so I'm a little skeptical that this analysis is telling us much. It's not clear how the Paolucci method is used in the study.

L130 – Provide some information about the scale of mapping for each of these maps.

L143-144 – It is problematic to make such a general statement about all landslides based on an analysis of three watersheds in one location.

L154 – Crest-clustering is not dominant for Chi chi either, I'd estimate that more than half of the cells are blue.

L163 – Also true for Fig 3

L182-185 Clarify if these values were adjusted somehow for topographic amplification, as described earlier, or if this is just showing the values directly reported by ShakeMap

L189 – At what DEM scale is the MAF computed? This definitely could benefit from more clarity earlier on how the MAF was actually computed, (i.e., at what scale) and then presumably averaged over macrocells. Wouldn't averaging it over such large areas tend to remove any possible correlations?

L230 – What is a dissolution figure?

L249 – Do the authors mean the landslides occurred when slopes were parallel to the

stratigraphic dip? That is what is implied by the cited figure.

Supplement

It was so frustrating to try to follow the supplement given its lack of clarity and poor English usage that I did not even try to provide comprehensive comments for it. The whole thing needs to be rewritten for clarity before it can be reviewed for scientific content. Some of the figures were hard to follow as well, especially Figure S8, which looks like a headless stick figure. It needs something for reference.

---

## Referee Comment (RC2) · Anonymous Referee #2 · 20 Mar 2019

In this paper, Rault and coauthors present a new statistical analysis of landslide position on hillslopes in four case studies: three coseismic and one rainfall-induced landslide event. From this analysis, the authors conclude that coseismic landslides do exhibit patterns grossly consistent with crest clustering, but that the distributions are best explained by a combination of both local geologic and seismic parameters. They further conclude that the typhoon example, Marakot in Taiwan, confirms that storm-induced landslides exhibit toe clustering.

The results are certainly interesting and worthy of publication. It's good to read a paper diving deeper into spatial patterns in landslides, especially using well inventoried case studies. I do, however, share many of the same concerns as Reviewer 1. Chief among these are the lack of explanation of the statistical methods employed in the analysis. I

was unable to fully understand the methodology after reading through either the main text or the supplemental material. In revision, I suggest adding more information to the main text, as well as expanding/editing the supplement, to make it accessible to an audience unfamiliar with the particular statistical methods used in this paper.

The second main issue that I have relates to conclusions from the typhoon dataset. It's hard to justify making sweeping, general claims from the study of one storm-induced landslide event. I think the result of toe clustering, in this particular case study, is compelling and that the spatial distribution should certainly be discussed. I would, in doing so, resist the urge to extrapolate the results to all storm-induced landslides. For comparison, consider the variability discovered among the three coseismic study sites. This larger dataset allows the authors to go into detail about the different geologic and seismic circumstances that aligned to drive those spatial patterns. Now imagine if the authors had studied just one of those examples and then generalized the result. It would be an inaccurate representation of the variety in distribution that they actually found when comparing multiple sites. So, in the case of the storm-induced landslides, I feel one is too small of a sample set to draw meaningful conclusions universal to rain-induced slides.

In addition to these main points, I also add the following comments/edits:

Line 41 – "1,2 to 2.5" the use of commas versus periods is inconsistent throughout the manuscript

Line 75 – Xu et al., 2014b is not in the references, this should probably be Xu et al., 2014

Line 78 – typo with a period before 31.9.

Line 143 – the word confirm feels too strong here (see main comment number 2 above)

Line 156 – I don't follow what is meant by this statement. Do the combined three case study site really show this, I thought you just described many differences between the

sites. Also, should be show not shows.

Line 163 – 174 – I suggest moving this section about scars vs. deposits to the supplemental material. It feels out of place and unnecessary at this position in the main text. Line 265 – So does this mean that the toe clustering in this case is a coincidence based on the position of weak rocks and/or faults? In other words, if weak busted up rocks or faults crossed through the middle of hillslopes (rather than toes) would you see more slides concentrated there or is there something particular about the toes?

Line 274 – typo – is should be in

Line 278 – what is meant by "may be revealed", is this meant to be a future study?

Line 457 – Xu reference should start on the next line

Figure 2 caption – "the black curve" should be the black line

Figure 5 caption – Is a word missing after Sichuan? the Sichuan what? Line 493 – I do not understand this sentence.

Figure 8 – where are the snapshot locations on this map? Am I missing a small box showing location(s)??

Supplementary Materials: Text has many typos, misspellings.

---

## Author Comment (AC1) · 15 Apr 2019

We would like to thank the referee for his review of the paper, helping us to improve the manuscript. The four main remarks are addressed first, followed by our responses to more detailed comments.

CR1 : It was very difficult to follow the methods description, which is mainly detailed in the supplement, because of English usage problems and misspellings, but also lack of clarity and logical gaps that require more explanation.

Response: We have completely rewritten the method, including more figures and a clear justification of its use. As it is more detailed than the previous version, we choose to leave it in the supplementary material to avoid overloading the main text with techni-

cal considerations. We hope that this new version is clearer.

CR1 : The authors state several times that they confirmed rainfall-induced landslides cluster at toes and earthquake-induced slides often cluster at crests in general in their analysis, but this statement is made based on comparing against just one rainfall inventory. That is not enough evidence to make such a strong, generalized statement. I'm quite sure there are other rainfall induced landslide inventories out there, adding one or two more to the analysis would provide more support to the general statements made.

Response: Indeed, in the Chi-Chi epicentral area, we compare the Chichi EQ-induced landslides inventory with the Typhoon Morakot inventory but also with the rainfall-induced landslides inventory covering the period from 1997 to 2013 (Cf figure 1 and table 1). Furthermore, we would like to highlight the fact that this observation is not only from our study but also observed by Densmore and Hovius, 2000 for storm-triggered catalogs in Idaho and California, in Taiwan for the Typhoon Toraji and Herb and for New Zealand storms (Meunier et al, 2008). We have clarified the main text. Changes: L153 "This evolution seems to confirm that landslides triggered by earthquakes and rainfall have distinct and different clustering behavior as observed in previous studies (Meunier et al, 2008; Densmore and Hovius, 2000)." L 172 "This observation, added to the results concerning the temporal variation of Rpcrest presented in the section 4.1, suggests that toe-clustering is a signature of rainfall-induced landslides."

CR1: The implementation of the topographic amplification estimation method needs to be explained in much greater detail, for it's not clear what they actually did to compute MAF, especially given that the method is frequency-dependent but they make no mention of assumptions of S-wave velocities or how they dealt with the frequency-dependence or how they chose the wavelength range they mention. But from what I can understand, I'm not sure their exploration of MAF as an explanatory factor for ridge clustering is not really telling us much, given the huge uncertainties in estimating topographic amplification, that they average the results over the very large macrocells,

and the fact that the method they use is based on numerical models and has not really been proven effective with real data yet. Note, there is an alternative empirical method that is not mentioned in the paper that might be worth comparing because in contrast to the Maufroy method, it is based on data. The downside is its specific to California data. Rai et al., 2016 https://doi.org/10.1193/113014EQS202M.

Response: We agree with the referee on this point. The MAF averaged in a macro-cell doesn't tell us more than what the concentrations of the ridges does. Therefore we have removed this part. We now refer to it only in the discussion, for area where topographic amplification might be an explanation for landslides position. However, in order to illustrate the fact that high ground motion at the top of ridges may explain the landslides position, as an example, we choose to show the soothed curvature computed for a wavelength similar to ridge sizes (i.e. That should correspond to different frequencies as the shear wave velocity varies spatially) in figure S8. As ridge curvature and relative elevation have a positive covariance (Rai et al, 2016 and figure S17), both indicators may be used to detect amplification at the top of the ridges.

Changes: All concerning the MAF have been removed from the main text. Figures 5 and S15 have been removed. Figure S17 have been added instead. We only suggest topographic amplification L270 "In these particular geological configurations, topographic amplification could control the landslide position. For example, in the Tangwanzhai syncline, the sharpest crests are oversampled by landslides (see supplementary topographic amplification, Fig. S17). Several authors have shown that ridge sharpness promotes topographic amplification (Maufroy et al., 2015; Rai et al., 2016)."

CR1: I'm not convinced they did a thorough job of ruling out bias because they derived the source areas from the original polygons automatically based on a simple assumption of typical aspect ratio and only looked at one inventory. I would be more convinced if they instead used one of the several inventories that did map source areas and deposits separately [e.g., Gorkha (Roback et al., 2017), Mid-Niigata (GSI of Japan, 2005), Kaikoura, (Massey et al.,2018)] to show whether considering entire polygons

biases the results.

Response: More catalogs would be better indeed but as far as we know, 1- the landslide-scar inventory of the Kaikura Earthquake is not available in open source 2- The Mid Niigata earthquake did not produce enough landslides to perform such analysis and 3- we have reasons to think that many of the landslide scars of the Gorkha inventory may be underestimated. We would like to draw the referee's attention on the fact that inventories presenting distinct scars can suffer from miss-mapping and are therefore subject to criticism. Instead, we choose to extract the scars using an empirical relationship relating the scar area to the total landslide area derived from a limited number of well documented cases. The same approach is used to derive landslide volumes from their surface. Since the patterns we observe using the landslide scar, the total landslide surface and the landslide centroid remain the same, we are quite confident in the fact that they are relatively robust. This means that the spatial variation of the landslide position remains the same. Moreover, this method can be applied to all landslide inventories mapping source areas and deposits altogether and so far, they remain the majority.

CR1: L13 – The word "confirms" is a little strong for a conclusion based on comparison against one rainfall inventory

Response: We have changed the sentence and added the fact that our observations agree with previous studies. See the answer to the second comment. Changes : L13 "A cross check against rainfall-induced landslide inventories seems to confirm that crest-clustering is specific to seismic-triggering as observed in previous studies"

CR1: L14-15 – Stating that seismic ground parameters have little bearing on observed patterns is pretty problematic because the landslides wouldn't have happened without the ground motion. The greater likelihood is that we don't have the means to accurately estimate the relevant ground motion parameters at the site where landslides are often triggered. Perhaps rather than saying "have little bearing" one could instead say ground

motion parameters from ShakeMap do not seem to exert a primary control on observed clustering patterns.

Response: We agree with the referee and have clarified this point. Changes: L15 "In our three study areas, the seismic ground motion parameters, lithological and topographic features used do not seem to exert a primary control"

CR1: L16-17 By major faults, do the authors mean faults involved in the earthquake that triggered the landslides or all faults?

Response: We mean "regional major faults", we precise it in the reviewed manuscript. Changes: L18: "Toe-clustering of seismically-induced landslides tends to occur along regional major faults."

CR1: L20-21 I don't think anyone is suggesting that landslide clustering be used as an indicator of seismic parameters

Response: We agree with the referee and have clarified this point. Changes: L18: "As a result the observation of landslide clustering on topographic ridges cannot be used as a definite indicator of topographic ground shaking amplification".

CR1: L61 – give reference/source of reported PGA's here and elsewhere. Use commas for thousands in English, dots for decimals.

Changes: L68 "(Tsai et al, 2000) ". L470 : "Tsai, Y. B., & Huang, M. W. (2000). Strong ground motion characteristics of the chichi, Taiwan, earthquake of September 21, 1999. Institute of Geophysics, National Central University. "

CR1: L95 – What does a random draw of landslide positions with no external forcing even mean? Pretty much every landslide occurs due to external forcing. Perhaps this section needs to be rewritten for clarity?

Response: It is not very clear indeed. We mean a set of landslides randomly drawn in the landscape so there is no bias (or forcing) on their position. It's the null hypothesis

(with regards to landslide position) we use to quantify the statistical robustness of any bias (clusters) observed in the data. Changes: All the supplementary and the main text has been rewritten for clarification.

CR1: L121 – How is the Maufroy method actually implemented? Not nearly enough detail is given. The method is frequency-specific, but the relevant frequency depends on the scale of the feature and the wavelength depends on the shear wave velocity.

Response: See the response to the third remark.

CR1: Also how is this applied to the ground motions, none of the ShakeMap outputs are frequency specific except the spectral accelerations, but those are single degree of freedom oscillators with a specific natural frequency, which is not the same thing as ground motion of a specific frequency content. This method is also based on modeling results and to my knowledge, hasn't yet really been validated against real data so I'm a little skeptical that this analysis is telling us much. It's not clear how the Paolucci method is used in the study.

Response: We have not combined the MAF with the ShakeMap. We have not used the Paolucci method in that paper. We have just mentioned its work as an example of a study showing possible relations between the topography and ground motion. The ridge width can be related to frequency of resonance of the topography (e.g. Paolucci, 2002, Massa et al, 2014) and the ridge shape ratio can be linked to the ground motion amplification (Geli, 1988).

Changes: L128 to 133 "For example, the ridge half width can be related to the frequency of resonance of the topography (e.g. Paolucci, 2002, Massa et al, 2014) and the ridge shape ratio (slope height /ridge width) can be linked to the ground motion amplification (Geli, 1988). To test if the clustering can be associated to the geometry of the ridges we calculate and associate to each macrocell the median slope heights and the median of the ridge half -widths (Fig. S8)."

CR1: L130 – Provide some information about the scale of mapping for each of these maps.

Response: We have reworked the section 3.3 of the main manuscript and added two figures to the supplementary material to address this point. Changes: L116-124: "Maps of Rpcrest and Rptoe were generated by subdividing a study area into macrocells in which Rp is calculated. The size of the macrocells in this study is set at 7.8 km2 to optimize for two criteria: a) the cell must be small enough to capture the spatial variation within the epicentral area, and b) it must be large enough to be statistically representative in terms of landslide content (see supplementary Methods-Metrics). The second criterion imposes a lower limit to the resolution at which we can observe any spatial variation. Figure S5 shows three Rpcrest maps in the Wenchuan epicentral area with increasing macrocell size. Although the patterns remain globally the same, macrocells of 7.8 km2 produce the most legible map. The mean of Rpcrest, averaged over the whole landscape, remains relatively independent of the macrocell size (Table 2, supplementary)."

CR1 : L143-144 – It is problematic to make such a general statement about all landslides based on an analysis of three watersheds in one location.

Response: Beside the clear signal we observe in these three watersheds, the Morakot landslides do cluster downslope over more than 1200km2. This pattern significantly differs from the three patterns associated to EQ-triggered landslides and we find this quite convincing. Moreover, the conclusion is not only made on our observations but also based on previous studies by Densmore and Hovius, 2000 and Meunier et al, 2008. Changes: L144 "This evolution seems to confirm that landslides triggered by earthquakes and rainfall have distinct and different clustering behavior as observed in previous study (Meunier et al, 2008; Densmore and Hovius, 2000)"

CR1: L154 – Crest-clustering is not dominant for Chi chi either, I'd estimate that more than half of the cells are blue. Changes: L158 we have removed "in contrast to the

other two cases"

CR1: L163 – Also true for Fig 3

Response: Fig 3 represents the clustering map of landslides triggered by the typhoon Morakot, for which we observe only toe-clustering.

CR1: L182-185 Clarify if these values were adjusted somehow for topographic amplification, as described earlier, or if this is just showing the values directly reported by ShakeMap.

Response: We only used the values from the ShakeMap, no adjustment with any amplification algorithms have been done, as the ShakeMaps already take in account in some way the topography. Changes: L189 "published on ShakeMap"

CR1: L189 – At what DEM scale is the MAF computed? This definitely could benefit from more clarity earlier on how the MAF was actually computed, (i.e., at what scale) and then presumably averaged over macrocells. Wouldn't averaging it over such large areas tend to remove any possible correlations? Response: All the part concerning the MAF has been removed.

CR1: L230 – What is a dissolution figure?

Response: We use the wrong expression, we mean high pressure solution evidences. Changes: L226 "rock layers are dipping steeply and bear traces of strong deformation, including pervasive schistosity (Robert, 2011)"

R1: L249 – Do the authors mean the landslides occurred when slopes were parallel to the stratigraphic dip? That is what is implied by the cited figure.

Response: In Northridge the landslides occurred on top of the scarp slopes. The legend of the figure S14 and the main text have been clarified (see L245). Changes: L245 "There, co-seismic landslides preferentially occurred on the top of the scarp slopes steepest slopes cutting across the stratigraphic dip"

Please also note the supplement to this comment:
https://www.earth-surf-dynam-discuss.net/esurf-2018-82/esurf-2018-82-AC1-
supplement.pdf

---

## Author Comment (AC2) · 15 Apr 2019

We thank the referee for his useful comments. We address the two main comments first and then the more detailed remarks.

R2 : Chief among these are the lack of explanation of the statistical methods employed in the analysis. I was unable to fully understand the methodology after reading through either the main text or the supplemental material. In revision, I suggest adding more information to the main text, as well as expanding/editing the supplement, to make it accessible to an audience unfamiliar with the particular statistical methods used in this paper.

Response: This point has also been raised by the first review so we have completely

rewritten the method, including more figures and a clear justification of its use. We choose to leave it in the supplementary material to avoid overloading the main text with technical considerations. We hope that this new version is clearer.

R2 : The second main issue that I have relates to conclusions from the typhoon dataset. It's hard to justify making sweeping, general claims from the study of one storm-induced landslide event. I think the result of toe clustering, in this particular case study, is compelling and that the spatial distribution should certainly be discussed. I would, in doing so, resist the urge to extrapolate the results to all storm-induced land-slides. For comparison, consider the variability discovered among the three coseismic study sites. This larger dataset allows the authors to go into detail about the different geologic and seismic circumstances that aligned to drive those spatial patterns. Now imagine if the authors had studied just one of those examples and then generalized the result. It would be an inaccurate representation of the variety in distribution that they actually found when comparing multiple sites. So, in the case of the storm-induced landslides, I feel one is too small of a sample set to draw meaningful conclusions universal to rain-induced slides. In addition to these main points, I also add the following comments/edits:

Response: We were not clear on that point as the 1st referee also pointed out. Actually, we do not base our conclusion from the Morakot dataset only. We also consider the time variation of the crest clustering (RP crest) derived from different rainfall-induced landslide inventories spreading over 16 years (fig. 1 and table 1). Technically, one could argue it constitutes 6 different dataset and both referees consider it being one just because we choose to merge them into one figure. Moreover, we clearly refer to the results of Densmore and Hovius 2000 and Meunier et al., 2007 to build up our conclusion.

Changes : L153 "This evolution seems to confirm that landslides triggered by earthquakes and rainfall have distinct and different clustering behaviour as observed in previous study (Meunier et al, 2008; Densmore and Hovius, 2000)." L 172 "This observa-

**ESurfD**
tion, added to the results concerning the temporal variation of Rpcrest presented in the section 4.1, suggests that toe-clustering is a signature of rainfall-induced landslides."

R2 : Line 41 – "1,2 to 2.5" the use of commas versus periods is inconsistent throughout the manuscript

Changes : We have changed comma for dots for decimals.

R2 : Line 75 – Xu et al., 2014b is not in the references, this should probably be Xu et al.,2014

Change done

R2 : Line 78 – typo with a period before 31.9.

Change done

R2 : Line 143 – the word confirm feels too strong here (see main comment number 2 above)

Change done

R2 : Line 156 – I don't follow what is meant by this statement. Do the combined three case study site really show this, I thought you just described many differences between the sites. Also, should be show not shows.

Response:.The 3 study cases show patterns of crest and toe clustering of landslides induced by earthquakes. Therefore, all show similar behavior even if their spatial extension strongly differs.

Changes : "Therefore, the three cases show ..."

R2 : Line 163 – 174 – I suggest moving this section about scars vs. deposits to the supplemental material. It feels out of place and unnecessary at this position in the main text.

Changes : L163 toL174 have been removed from the main text and added to the

supplementary

R2: Line 265 – So does this mean that the toe clustering in this case is a coincidence based on the position of weak rocks and/or faults? In other words, if weak busted up rocks or faults crossed through the middle of hillslopes (rather than toes) would you see more slides concentrated there or is there something particular about the toes?

Response: We are talking about major regional faults. In the case of the Sichuan, some major rivers (such as the Minjang river) flow along them. Therefore the most weakened rocks are located downslope in those valleys, producing toe clustering. So yes, it's the results of the rivers lining up with major fault zones.

R2 : Line 274 – typo – is should be in

Change done

R2 : Line 278 – what is meant by "may be revealed", is this meant to be a future study?

Changes: This sentence has been removed. Instead we refer the reader to the supplementary.

R2 : Line 457 – Xu reference should start on the next line

Change done

R2 : Figure 2 caption – "the black curve" should be the black line

Change done

R2 : Figure 5 caption – Is a word missing after Sichuan? the Sichuan what? Line 493 – I do not understand this sentence.

Changes: The figure 5 has been removed

R2 : Figure 8 – where are the snapshot locations on this map? Am I missing a small box showing location(s)??

Changes: Figure 8 has been changed. Many of the snapshots have been removed, the small boxes of those represented have been added. Legend: "Figure 7: a. Structural map of the Wenchuan earthquake epicentral area (after Robert, 2011) overlaid with the Rpcrest map. b. Snapshot of the landslide map in a portion of the Wenchuan shear zone. Its location is reported in Fig. 7a. Polygons with red contours represent the co-seismic landslides mapped by Xu et al., 2014. The yellow and blue lines delimit zones of crest- and toe- clustering respectively. c. Cross sections of different structural units after Robert 2011. Cross sections I-J and K-L are presented in Fig. S13. GF: Guanxian fault, BF Beichuan fault, WF Wenchuan fault, Y-B F Yinxiu-Beichuan fault, QF Qinling fault."

R2 : Supplementary Materials: Text has many typos, misspellings.

Corrected

Please also note the supplement to this comment:
https://www.earth-surf-dynam-discuss.net/esurf-2018-82/esurf-2018-82-AC2-supplement.pdf

---

## Author Response (AR2)

**Responses to Reviewer 2 :**

We thank the referee for his feedbacks and corrections on our manuscript.  We have done the suggested technical corrections.

R2 : Line 234 - is a word missing in the phrase "and important fault system"?

Change : "and important fault system"? ➔ steep valleys, aligned major faults, cut shaly sandstones and slightly metamorphosed argilite layers.

R2 : Line 243 - "very steep" hillslope is vague, can you provide a slope angle range?

Change :  on the steepest hillslopes (>26°)

---

## Author Response (AR3)

We thank the editor for his useful remarks. We have made the appropriate changes to improve both the manuscript and the supplement according to the notes he gave. In the following, we give a point by point description of the changes we made.

(1) throughout the main text (e.g., lines 119, 153, etc), the reference to the supplement is inconsistently formatted and does not accurately reflect the subheadings in the supplement sections; please correct this to be consistent and accurate

We have reformatted all the references to the supplement in order to reflect the sections of the supplementary material.

(2) please double check to make sure that the figures are uploaded at high resolution and appear correctly in the final PDFs; in some cases, the versions in some of the drafts for review are unacceptably low resolution for final publication (please check the final figure resolution not only for the main text proofs, but also for the supplement which will not be reviewed in detail during copy editing) — actually the most recent version seems ok, but just check these

All the figures have been checked.

(3) A few minor line-by-line notes:
Lines 14-15: reword to "seismic ground motion parameters AND lithologic and topographic features"
Lines 218: the "." should be removed before the citation to Robert, 2011
Line 269: add a "." after "e.g"

Changes made.

In the Supplement
Line S2: subtitle should be "Methods and metrics" to follow formatting of other subtitles (not capital M on Metrics)
Lines S24,S33: the brackets designating the interval range here are opposite the format used in the main text (line 98)
Line S42: "The use of Rp TO remove..."

Changes made.

Lines S46-47: it seems odd wording to say that you want to insure that the landslides oversample a given hillslope portion — presumably you want to make sure that values for Rp>1 represent an oversampling — these are different things, and I think the wording could be improved

The sentence is now: "In other words we want to quantify the null hypothesis that $R_p>1$ (or $R_p<1$) is due to random fluctuations around $PDF_{topo}$ and hense insure that we retain only statistically robust cases (macrocells) of landslide clustering."

Line S69: probability should not be capitalized unless there is a reason I don't understand
Line S81: errant space before the first sentence here
Line S124: "THESE parameters"

Changes made.

Figure S1 caption: I think it would help to specify the flow accumulation threshold used for channel delineation for this landscape, and more generally in this study (maybe I missed this elsewhere); also, can you clarify which of the three ridge delineation methods were used in calculating Rp as reported in the main text?

Supplementary material now includes the following statements:

Line S21-S23: " The thresholds of drainage area we use to define channel heads vary from 0.02 to 0.5 km$^2$ in this study. Crests are mapped using a double criterion of null flow accumulation and a threshold of positive curvature (Fig. S1b)."

Fig. S1 caption: "Channel heads are evaluated after Montgomery, 2001, using a threshold of 0.07 km$^2$ of drainage area. Crests are generated with three different methods: Crest cells are mapped as a) cells of null flow accumulation (NFA) b) cells of NFA and above a positive curvature threshold (PCT) (used in this study) and…"

Figure S14 caption: It may be worth noting here that the extent of these areas is illustrated in Figure 7
Fig. S14 caption: "The locations of a. and b. are reported in Fig. 7a. "

The authors